# High correlation between detection of dengue IgG from dried blood spots and serum using an indirect IgG ELISA assay: A validation study in Fortaleza, Brazil

**Monica Zahreddine[1], Beatriz Parra[2], Laura Pierce[1], Danielle Ferreira de Oliveira[3], Mabel Carabali[1,4], Katia Charland[1], Kellyanne Abreu[5], Valéry Ridde[6], Danielle Malta Lima[3,7], Kate Zinszer[1,8]***

**1** Centre de recherche en santé publique, Université de Montréal, Montréal, Québec, Canada, **2** Microbiology Department, Universidad del Valle, Cali, Colombia, **3** Universidade de Fortaleza, Fortaleza, Brazil, **4** Department of Epidemiology, Biostatistics, and Occupational Health, McGill University, Montréal, Québec, Canada, **5** Universidade Estadual do Ceará, Ceará, Brazil, **6** Université Paris Cité, IRD, Inserm, Université Sorbonne Paris Nord, Ceped, Paris, France, **7** Graduate Program in Medical Sciences, Universidade de Fortaleza, Fortaleza, Brazil, **8** École de santé publique, Université de Montréal, Montréal, Québec, Canada

* kate.zinszer@umontreal.ca

## Abstract

### Background

Dengue virus (DENV) seroprevalence studies often rely on Enzyme-Linked Immunosorbent Assay (ELISA) testing of serum samples, but ELISA testing of dried blood spot (DBS) samples offer several advantages for field-based research in resource-limited settings. However, DBS' limited sample volume can be challenging for test sensitivity, requiring validation studies with standard methods (e.g., analysis of serum through ELISAs or Plaque Reduction Neutralization Tests (PRNTs)). In preparation for a large cluster randomized controlled trial, we conducted a pilot study in 2019 to validate the use of DBS compared to serum samples for DENV IgG testing. We aimed to identify the optimal DBS dilution for IgG detection and to estimate the correlation, magnitude of agreement, and sensitivity and specificity of IgG detection in DBS versus serum samples.

### Methodology/Principal Findings

We conducted this pilot validation study among 119 healthy participants in Fortaleza, Brazil to evaluate and optimize the detection of DENV IgG from DBS compared to serum. Each participant provided paired DBS and venous blood samples, which were evaluated for DENV IgG using the Panbio Dengue IgG indirect ELISA. DBS elution diluted 1:4 was optimal compared with serum results, with high correlation (r= 0.98) and near-perfect agreement (kappa = 0.95). At this dilution, DBS had a sensitivity of 100%, a specificity of 92.3%, a 97.9% positive predictive value, and a 100% negative predictive value compared with serum.

**Data availability statement:** All de-identified or anonymized data used to replicate this study's findings can be accessed through the research ethics board at Université de Montréal, CERSES (email: cerses@umontreal.ca) to ensure that data is shared in accordance with participant consent since data contain potentially sensitive personal health information.

**Funding:** This study was funded by the Canadian Institutes of Health Research (CIHR), through a project grant to Kate Zinszer (201803PJT-400444-RC2-CFCA-120159).

**Competing interests:** The authors have declared that no competing interests exist.

## Conclusions/Significance

These results validate using DBS instead of serum for detection of prior dengue infection among similar populations in endemic regions, without sacrificing test sensitivity and specificity. The validity of using DBS for ELISA to detect prior dengue infection could have important implications for field-based research. A limitation to this study was that the potential for misclassification due to cross-reactivity (e.g., with Zika virus, Yellow Fever vaccine) was not assessed.

## Author summary

Dengue is a viral disease spread by *Aedes* mosquitoes in tropical and subtropical regions with over 40% of the world's population considered at-risk of infection. Traditional methods of assessing prior dengue infection can be costly and invasive (e.g., blood draw) which are impractical for population-based surveillance or large field-based studies. Whole blood collected via a finger-prick is an alternative option for measuring past infection by identifying antibodies using serological methods. This study compared two methods of blood sample collection for the purpose of identifying previously infected individuals: a serum sample collected by venipuncture vs. a dried blood spot sample collected by finger-prick. Study nurses collected both types of samples from each research participant and they were analyzed in a laboratory for the presence of antibodies from a prior dengue infection. Results of the laboratory analyses between the two different types of samples were very similar. This suggests that the simple dried blood spot (DBS) approach can be used with reasonable validity in similar study populations to evaluate prior dengue infection.

## Introduction

Dengue is one of the most important neglected tropical diseases in the world, with almost 400 million people per year infected by the mosquito-borne virus [1]. There has been a significant increase in dengue over the past 50 years, with an estimated 30-fold increase in incidence [2,3]. Dengue is an acute febrile disease caused by four distinct but closely related serotypes of dengue virus (DENV), DENV-1, DENV-2, DENV-3, and DENV-4 [4,5].

Dengue can be diagnosed using antigen (e.g., Non-Structural protein 1, NS1), antibody (e.g., immunoglobulins M or G, IgM or IgG), or molecular testing (e.g., polymerase chain reaction) [6–10]. Seroprevalence studies commonly use DENV IgG antibodies to identify previous dengue infection, whereas DENV IgM antibodies or a four-fold increase in IgG antibody titers are used to identify acute infections. Currently, there are several diagnostic tests available for detecting DENV IgG antibodies. Among them, DENV-IgG Enzyme-Linked Immunosorbent Assays (ELISAs) are used extensively for serosurveys as they are less costly and time-consuming than Plaque Reduction Neutralization Tests (PRNTs), although ELISAs are less sensitive and specific compared to PRNTs [11,12]. In laboratory settings, serum is the standard specimen for DENV IgG measurements by ELISA, but in field-based research settings, the use of serum poses a high burden to the participant, particularly in children, because of the need for blood collection by venipuncture. The costs and logistics associated with venipuncture are also major barriers, particularly in remote, low-resource settings with limited infrastructure [8]. An alternative to venipuncture is a finger prick to collect drops of capillary

whole blood on filter paper. These dried blood spots (DBS) are minimally invasive, low cost, and do not have cold chain requirements for transport and temporary storage as DBS can be stored at room temperature and are reconstituted with a simple elution step. Despite being non-standard specimens and having a small-blood volume, DBS advantages usually outweigh their disadvantages in field-based serosurveys [13–15].

In 2019, a cluster randomized controlled trial (cRCT) was initiated in Fortaleza, Brazil to determine if interventions based upon community mobilization reduce the risk of DENV infection in children compared to usual dengue control practice [16]. DBS was chosen as the sampling method to reduce sampling burden on children and to facilitate sample collection and transportation in a community-based research setting. Prior to the serological analysis of DBS samples from the cRCT, a pilot study was conducted to evaluate and optimize the detection of DENV IgG from DBS using the dengue IgG indirect ELISA assay. Specifically, our objective was to define the optimal DBS dilution for IgG detection, compared to serum and assess the concordance between the DBS and serum classification of dengue IgG.

## Methods

The study design was a cross-sectional study to evaluate the detection of DENV IgG antibodies using an elution technique with DBS samples compared to the current standard using serum samples.

### Ethics statement

The study protocol was reviewed and approved by the University of Montreal's Comité d'éthique de la recherche en sciences et en santé (CERSES, Canada, reference number 18-141), and Comité de Ética em Pesquisa da Universidade Estadual do Ceará (State University of Ceará, Brazil, reference number 3.083.892). Written informed consent was obtained from all participants.

### Eligibility criteria and sample collection

To evaluate our DBS method and assay, we required both IgG DENV naive/susceptible (i.e., never infected) participants and IgG DENV-positive (i.e., previously infected) participants. Individuals were eligible for participation if they were current students and attended the Universidade Estadual do Ceará (UECE) in Fortaleza, Brazil, and if they did not present any acute symptoms (e.g., fever, headache) at the time of enrollment. A brief health history and symptom questionnaire was administered to ensure the exclusion of symptomatic individuals. All participants were called from research personnel with their test results with an accompanying interpretation of the results.

Research nurses at the University campus collected a serum sample and a capillary DBS sample from each participant. Approximately 5-6 mL of blood was collected by venipuncture from each participant into a BD Vacutainer clot-activator gel tube. For the DBS sample, a Contact-Activated Lancet (BD Microtainer) was used to collect capillary blood from the fingertip on filter paper (Whatman 903). Each circle of a Whatman 903 filter paper card holds 75 to 80 μL of sample. Three half-inch (12.7 mm) circles were filled with whole blood and air dried at room temperature. Once dry, each card was individually packaged in a hermetic plastic bag containing silica desiccant. After collection, both samples were transported in refrigerated containers to the Pathogenic Bioagents Research Laboratory at the University of Fortaleza. Venous blood samples were centrifuged (Hettich/RotoFix32A) at 3500 rpm for 10 minutes to obtain serum, and subsequently aliquoted into cryogenic tubes and stored in a freezer at –20°C. DBS samples were visually inspected for quality control purposes, in terms

of insufficient quantity of blood or layering of blood drops, since these would be disqualified for use due to an inability to estimate the sample volume. The DBS samples were then stored inside their individual bags in a freezer at -20°C.

## Measurement of DENV IgG antibodies

Samples from DBS filter paper were eluted as described in the literature [17,18]. Briefly, three DBS punches of 6 mm were soaked in a 1.5 mL vial containing 250 µL of eluent solution [PBS (sodium phosphate buffer pH 7.2), 0.5% bovine serum albumin (Sigma-Aldrich) and 0.3% Tween 20], shaken at 300 rpm overnight at 4°C to ensure adequate serum elution. The next day, the eluates were centrifuged at 2500 rpm for 2 minutes in a refrigerated microfuge and then transferred to a sterile tube. Undiluted DBS eluates were treated as a 1:25 equivalent dilution of human serum, as three 6 mm punches were estimated to contain 21 µL of dried blood, corresponding to approximately 10.5 µL of serum that were diluted in 250 µL of the diluent. DBS eluates were tested undiluted and diluted to 1:2 and 1:4, to equate to 1:25, 1:50 and 1:100 dilutions of the serum sample, respectively. Serum samples were diluted to 1:100, according to the manufacturer instructions.

The eluates were immediately used for qualitative detection of IgG antibodies against DENV-1 to 4 using the Panbio Dengue IgG indirect ELISA test (Panbio diagnostic, Alere/ Standard Diagnostics, INC., Republic of Korea) along with the matched serum sample, following the manufacturer recommendations [19]. Briefly, 100 µL of each thawed serum sample diluted to 1:100, its equivalent DBS eluate (diluted 1:4), as well as DBS eluate undiluted and diluted 1:2, were transferred to the coated microwells and incubated for 30 minutes at 37°C. Each plate was washed 6 times. Horseradish peroxidase (HRP) conjugated anti-human IgG was added to each well and plates were incubated for an additional 30 minutes. After a second washing, tetramethylbenzidine was added to each well and incubated for 10 minutes at room temperature, followed by the addition of the stop solution. The absorbance of each well was read using a microplate reader. The calibrator was included in each plate in triplicate, including one negative control and one positive control provided by the kit.

## Sample size calculation

A sample size calculation was performed, assuming a correlation of 85% between DBS and serum for dengue IgG detection with a maximum discrepancy of 10% between the two methods, a power of 80% and an alpha (error rate) of 5%. This resulted in a suggested sample size of 101 participants with 2 samples per person. We aimed to screen 112 potential participants, assuming that 10% of participants would not be eligible. The estimated sample size was suitable for different scenarios of sensitivity (75-85%), specificity (85-90%) and error rate 10% of the diagnostic test, accounting for a DENV seroprevalence between 30-40% in the study area [20].

## Assessing concordance of DBS and serum classification of dengue IgG

As stated in ELISA manufacturer's instructions, DBS and serum samples presenting an index value > 1.1 were considered positive, and those in the grey area from 0.9 to 1.1 were defined as equivocal and classified as seronegative for analysis purposes according to a-priori decision. Scatterplots were created comparing DENV IgG index values of serum samples with their paired DBS sample, and a correlation coefficient was estimated for each DBS dilution tested (non-diluted, diluted to 1:2, and diluted to 1:4). The identification of the optimal DBS dilution was based on the DBS index value that most closely resembled the index value, or the result (positive or negative), obtained with the matched serum sample. Then, we estimated the sensitivity, specificity, positive predictive value (PPV), and negative predictive value (NPV) of DBS

at each dilution, compared to serum. The differences between the index values for matched DBS and serum samples (DBS index value – serum index value) were plotted against their mean ((DBS index value + serum index value)/2) following the Bland-Altman method [21,22]. The upper and lower limits of agreement (LOA) were identified as well as 95% confidence intervals (95% CIs).

Low, intermediate, and high positive ranges were estimated based on the terciles of our sampling pool, and were defined as 1.1 to < 1.7, 1.7 to < 2.1, and ≥ 2.1, respectively. From the 119 participants, 30 positive results were retested using the defined optimal DBS dilution, with 10 results retested from each tercile (low, intermediate, high) of index values. Four replicates of the DBS sample per participant were retested in the same plate using the optimal dilution. Another two replicates were retested in a different plate using the same dilution. Intra-assay and inter-assay coefficients of variation (COV) were calculated for each tercile of positive index values. Intra-assay COV is the average coefficient of variation between replicates, whereas inter-assay COV is the average coefficient of variation between plates. To calculate the intra-assay COV of each tercile, a COV was first calculated for each of the 10 samples individually by taking the standard deviation of the four replicates' index values, dividing by the mean of the four replicates' index values, and multiplying by 100. The intra-assay COV was then calculated as the mean of the 10 sample COVs. To calculate the inter-assay COV, the mean of mean index values was calculated for each tercile and plate (i.e., the mean of the four replicate index values of the 10 samples was calculated and averaged). Then, the inter-assay COV was calculated as the standard deviation of the mean of means from the two plates, divided by the average of the mean of means from the two plates.

## Results

One hundred and nineteen healthy individuals were recruited and enrolled at the UECE campus in December 2019. The mean age of participants was 22.3 years (standard deviation [SD] 5.6), and most were female. Thirty-five (29%) participants reported having a previous episode of dengue, and 13 (11%) of Zika. Twenty-eight (24%) recalled receiving a previous Yellow Fever immunization (Table 1).

All 119 participants provided matched serum and DBS samples, which all passed the laboratory quality control standards. Ninety-three (78.2%) of the serum samples were seropositive compared to 95 (79.8%) DBS samples diluted to 1:4 (Table 2). Equivocal samples were classified as seronegative for analysis purposes. Approximately equal proportions of individuals reporting either a previous Zika episode or a Yellow Fever vaccination were seropositive and seronegative for DENV, respectively (Table 2).

DBS elution diluted 1:4 was found to be the optimal sample dilution, compared with the results of the matching serum diluted according to the manufacturer instructions (1:100) (Figs 1–3). IgG detection using serum and DBS with 1:4 dilution showed a very high correlation (r= 0.98) and near-perfect agreement (kappa = 0.95). DBS elution diluted 1:4 had a sensitivity of 100%, a specificity of 92.3%, a 97.9% PPV, and a 100% NPV compared with serum (Table 3).

The Bland-Altman mean of differences between the index values of DBS diluted 1:4 and the matched serum samples was 0.041 (95% CI: 0.004 – 0.079), with upper and lower LOAs of 0.443 (95% CI: 0.380 – 0.507) and –0.361 (95% CI: –0.424 – –0.297), respectively (Fig 4). The index value of a given DBS sample was, on average, 0.041 units greater than the index value of the matched serum sample. The bias was greater for index values less than 1.5, as shown by the distribution in Fig 4.

Intra-assay COV were 4.57%, 3.73% and 5.93% for lower, intermediate, and higher ranges, respectively, all lower than the accepted limit of 10%. Inter-assay COV were 3.53%, 3.73%,

**Table 1. Study population characteristics (n=119).**

| Age (years) | |
| --- | --- |
| Median (IQR) | 20.8 (19.1–23.3) |
| | **n (%)** |
| **Sex at birth** | |
| Female | 75 (63.0) |
| Male | 44 (37.0) |
| **Chronic disease or health condition** | |
| No | 112 (94.1) |
| Yes | 7 (5.9) |
| **Using medication** | |
| No | 80 (67.2) |
| Yes | 39 (32.8) |
| **Presence of acute symptoms** | |
| No | 119 (100.0) |
| Yes | 0 (0.0) |
| **Symptoms in the last 4 weeks**[*] | |
| No | 98 (82.4) |
| Yes | 21 (17.6) |
| **Previous dengue episode** | |
| No | 77 (64.7) |
| Yes | 35 (29.4) |
| Does not know | 7 (5.9) |
| **Previous Zika episode** | |
| No | 102 (85.7) |
| Yes | 13 (10.9) |
| Does not know | 4 (3.4) |
| **Previous Yellow Fever vaccine** | |
| No | 78 (65.5) |
| Yes | 28 (23.5) |
| Does not know | 13 (10.9) |

[*]symptoms of fever, diarrhea, respiratory or other acute symptoms.

and 5.33% for lower, intermediate, and higher ranges, all lower than the accepted limit of 15% (Table 4).

## Discussion

We found a strong correlation between dengue IgG detected in DBS and serum using an indirect IgG ELISA among a sample of healthy participants in Fortaleza, Brazil. These results indicate that there is high sensitivity and specificity when using DBS, with a 1:4 dilution, as a substitute for serum. Seropositivity for DENV IgG was very high in our sample, measured at 78.2% via serum and 79.8% via DBS, though consistent with other recent studies conducted in urban settings of Brazil among varied populations [23–25].

There were only four instances of discordant results between matched serum and DBS samples. In two cases, matched samples were classified as negative in serum and equivocal in DBS, and were therefore analyzed as negative samples. Two samples classified as positive via DBS were classified as negative (n=1) and equivocal (n=1) via serum, respectively. The

**Table 2.  Classification of participant seropositivity according to certain characteristics.**

| | | Serum Positive (n=93) n (%) | Serum Negative (n=25) n (%) | Serum Equivocal (n=1) n (%) | Total n (%) |
|---|---|---|---|---|---|
| DBS seropositivity | Positive | 93 (98) | 1 (1) | 1 (1) | 95 (100) |
| | Negative | 0 (0) | 22 (100) | 0 (0) | 22 (100) |
| | Equivocal | 0 (0) | 2 (100) | 0 (0) | 2 (100) |
| Previous self-reported dengue episode | Yes | 32 (91) | 3 (9) | 0 (0) | 35 (100) |
| | No | 55 (72) | 21 (27) | 1 (1) | 77 (100) |
| | Does not know | 6 (86) | 1 (14) | 0 (0) | 7 (100) |
| Previous self-reported Zika episode | Yes | 11 (85) | 2 (15) | 0 (0) | 13 (100) |
| | No | 78 (76) | 23 (23) | 1 (1) | 102 (100) |
| | Does not know | 4 (100) | 0 (0) | 0 (0) | 4 (100) |
| Self-reported Yellow Fever vaccination | Yes | 22 (79) | 6 (21) | 0 (0) | 28 (100) |
| | No | 60 (77) | 17 (22) | 1 (1) | 78 (100) |
| | Does not know | 11 (85) | 2 (15) | 0 (0) | 13 (100) |
| Previous self-reported Dengue or Zika episode, or Yellow Fever vaccination | Yes | 53 (87) | 8 (13) | 0 (0) | 61 (100) |
| | No | 24 (60) | 15 (38) | 1 (2) | 40 (100) |
| | Does not know | 16 (89) | 2 (11) | 0 (0) | 18 (100) |

Bland-Altman plot indicated that at index values below 1.5, DBS samples tended to have a higher signal than the matched serum sample (Fig 4). While this did not lead to discordant results in our sample, other studies may need to use caution or validate the use of alternate DBS dilutions (e.g., 1:8) to avoid over-estimating seropositivity when many results are within or close to the grey area of index values 0.9 to 1.1. However, mean differences were more equally distributed at index values above 1.5, and LOAs were generally small, indicating good overall agreement (Fig 4). When positive results from each tercile were retested, the observed intra- and inter-assay COVs were all lower than 10% and well within accepted ranges of reliability, validating the consistency of the laboratory calibration and processing techniques.

The seroprevalence results in our sample indicated greater circulation of dengue than that captured by self-reported prior dengue episode, which is consistent with other studies demonstrating the underreporting of dengue in Brazil and Thailand [26,27]. We found that 29% of participants (n=35) reported having a previous dengue episode; 94% (n=33) of these participants were classified as seropositive according to their DBS result and 91% (n=32) according to their serum result, respectively. Among those who reported no prior dengue episode (or no knowledge of a prior episode), DENV seroprevalence was measured at 73% via serum and 74% DBS. While higher than our original estimates and conducted within a small sample, these results were consistent with studies among similar urban areas of Brazil, which estimated seroprevalence of 54% among children in Fortaleza [23], and between 74-91% for children and adults in Recife and São Paulo [24,25].

An inherent limitation of dengue serological testing is the possibility of overestimating seropositivity due to cross-reactivity issues. Cross-reactivity for DENV IgG can occur after Yellow Fever vaccination [28,29] and from previous infection with the Zika virus [30,31], which caused large outbreaks in Brazil in 2015-2016 [32,33]. The magnitude of cross-reactivity for DENV IgG ELISAs has been estimated as 3.9–15.1% for those with prior Yellow Fever vaccination [29] and 54.0% for those with a Zika infection, although this was based on a small sample [29,34]. As 11% of our sample reported a prior Zika episode and given the magnitude of Zika and DENV cross-reactivity, we cannot rule out the possibility of false positives

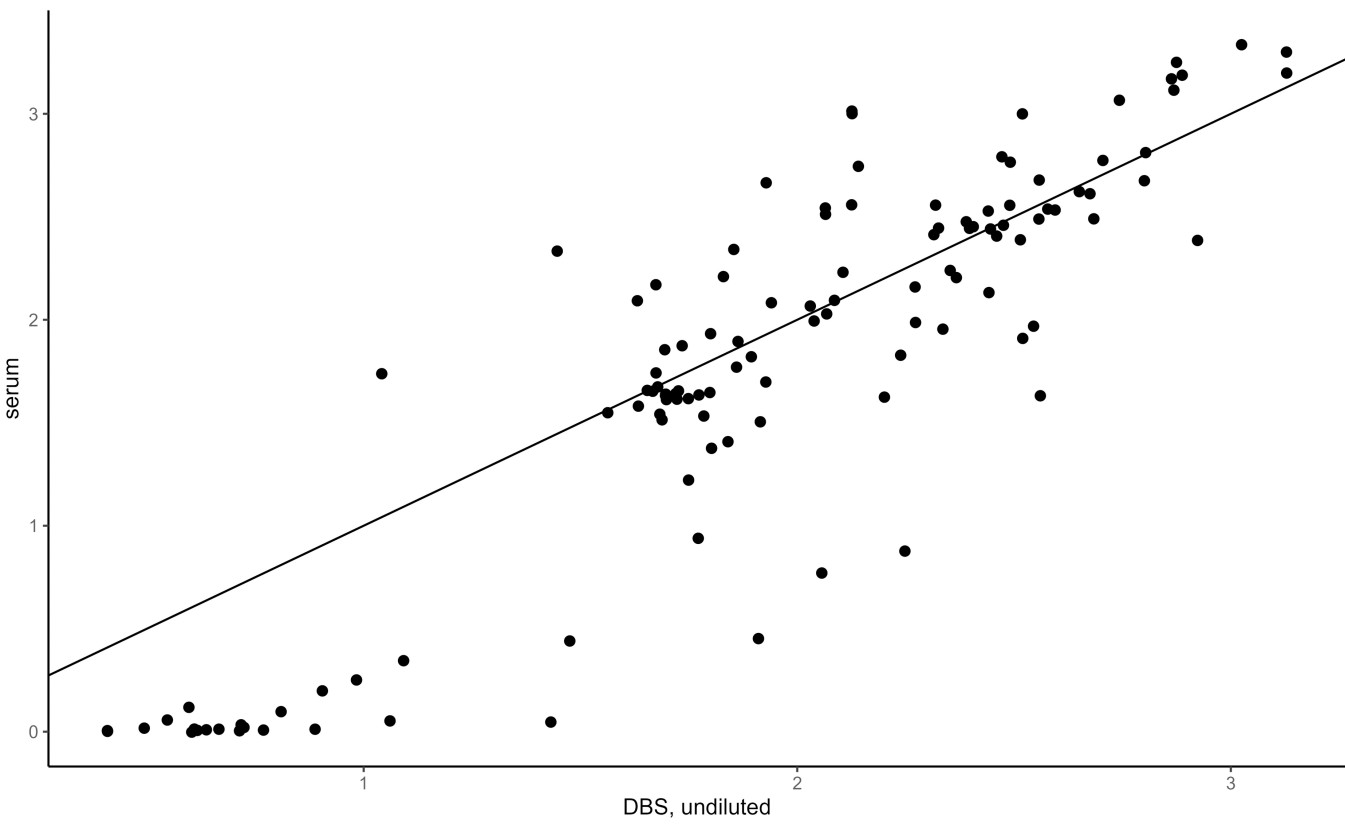

**Fig 1. Scatterplots of index values from non-diluted DBS eluates compared to serum samples diluted 1:100 (n =119 matched pairs of serum and DBS eluates).**

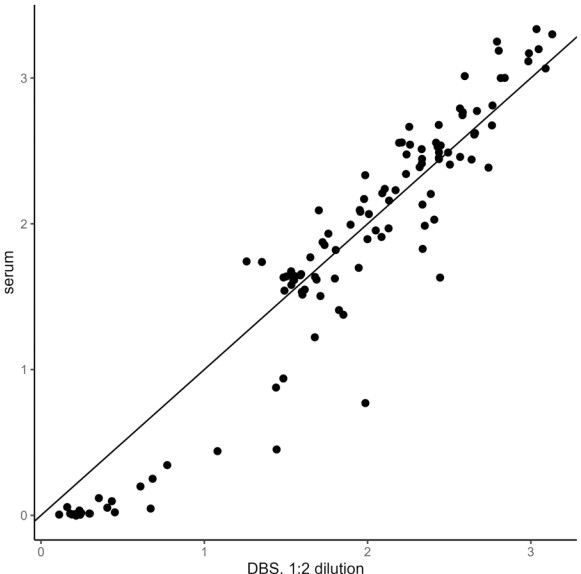

**Fig 2. Scatterplots of index values from DBS eluates diluted 1:2 compared to serum samples diluted 1:100 (n =119 matched pairs of serum and DBS eluates).**

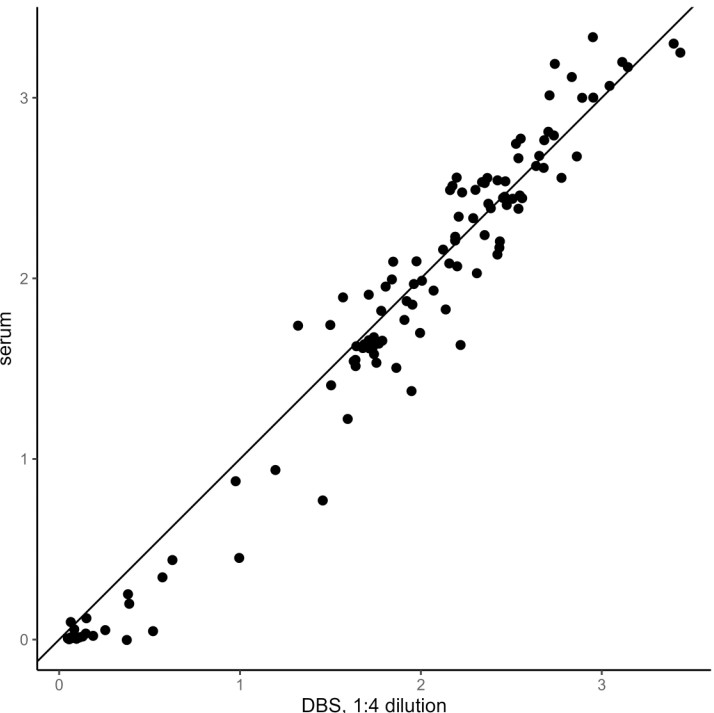

**Fig 3. Scatterplots of index values from DBS eluates diluted 1:4 compared to serum samples diluted 1:100 (n =119 matched pairs of serum and DBS eluates).**

**Table 3. Correlation, sensitivity, specificity, positive predictive value, and negative predictive value of DBS (n = 119 matched pairs of serum and DBS eluates).**

|  | DBS eluate sample | | |
|---|---|---|---|
|  | **Non-Diluted** | **Diluted to 1:2** | **Diluted to 1:4 (optimal dilution)** |
| Kappa | 0.815 | 0.896 | 0.949 |
| Z | 8.97 | 9.83 | 10.4 |
| p-value | <0.001 | <0.001 | <0.001 |
| Sensitivity | 0.99 | 1 | 1 |
| Specificity | 0.77 | 0.85 | 0.92 |
| Positive Predictive Value | 0.94 | 0.96 | 0.98 |
| Negative Predictive Value | 0.95 | 1 | 1 |
| Correlation Coefficient (r) | 0.90 | 0.96 | 0.98 |

among this subgroup of participants. The use of self-reported prior Zika diagnosis and Yellow Fever vaccination is another limitation; while data is limited on the accuracy of self-report for these outcomes, dengue self-report has been found to have poor sensitivity and only moderate specificity [27]. Future DBS validation studies should consider PRNT analysis or Zika ELISA for the samples of participants with evidence of both prior Zika and dengue infection. Though 24% of our sample reported receiving a prior Yellow Fever vaccine, the specificity of the Panbio Dengue IgG indirect ELISA has been estimated at 98.2% (95% CI 90.6–99.7) among Yellow Fever vaccinated individuals [35], and we found similar proportions of seropositive and seronegative participants reported receiving a prior Yellow Fever vaccine. While subgroup

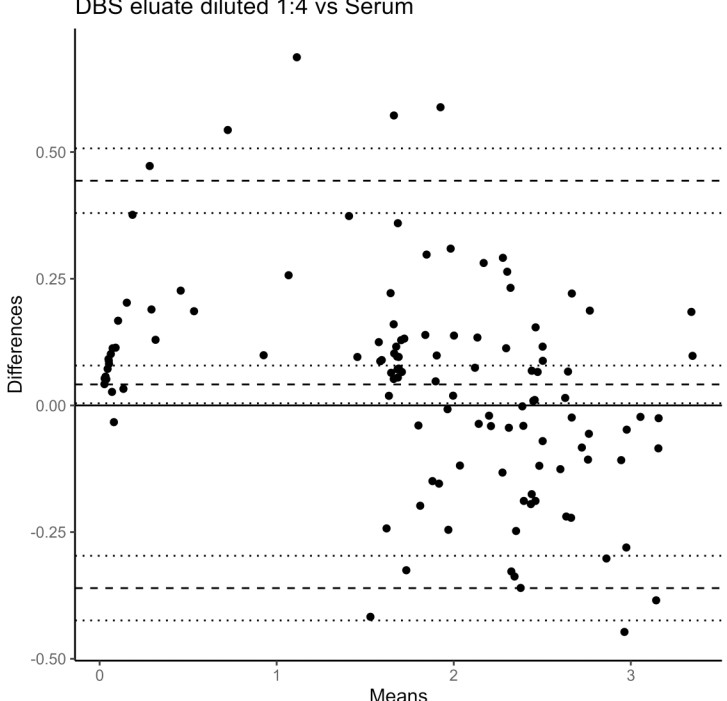

**Fig 4. Bland-Altman plot of the differences and means of dengue IgG indirect ELISA index values for serum samples and matched DBS samples at optimal dilution (1:4) (n = 119).** The central dashed line is the Bland-Altman mean of differences, with adjacent horizontal dotted lines indicating the corresponding 95% CI. The lower dashed line is the lower LOA, while the upper dashed line is the upper LOA. The 95% CI of each LOA is represented by adjacent horizontal dotted lines.

**Table 4. Intra-assay and Inter-assay coefficient of variation (COV) by positive index value terciles: low, intermediate, and high (n=30 positive results that were retested).**

|  | Intra-assay %COV | Inter-assay %COV |
|---|---|---|
| Low positive (n=10) | 4.57 | 3.53 |
| Intermediate positive (n=10) | 3.73 | 3.73 |
| High positive (n=10) | 5.93 | 5.33 |

sizes were small and insufficient for further analysis, there was no clear evidence of misclassification from cross-reactivity with a prior Yellow Fever vaccination.

The sensitivity of commercial serological tests to detect primary infections in serum via IgG-based ELISAs remains problematic for accurately measuring dengue seroprevalence in some populations. While the indirect IgG ELISA is an accepted and feasible technique for use in resource-limited settings and has been recommended by the WHO for pre-vaccination serosurveys [36], it has poor sensitivity to detect primary infections in serum [19,37]. In a primary infection, IgG may become detectable 10–15 days after illness onset, in which time diagnostic capabilities could be enhanced by concurrent assessment of more acute diagnostic markers such as NS1 antigen or IgM [7]. IgG-based ELISAs may underestimate seroprevalence and/or have more limited utility in settings with a high likelihood of cross-reactivity. While DBS performed well compared to serum using this assay and seroprevalence was high, we cannot rule out the possibility that some individuals with a primary DENV infection may

have been misclassified as seronegative by both serum and DBS. Therefore, when feasible, future validation studies should consider using micro-neutralization or plaque reduction neutralization assays as the gold standard comparator.

Another limitation of DBS testing is the qualitative assessment of seropositivity, without clear estimation of antibody concentrations [38]. The use of index values for quantitative validation purposes, as completed here, has been justified elsewhere [39], but several factors limit the further analysis of quantitative DBS results, including the possibility of variation in antibody concentrations (the hematocrit problem) among sampled individuals [39].

Finally, validation studies between DBS and other methods, such as ELISA of serum samples, remain important to assess the applicability of methods to new settings and populations. In the case of this study, our population was young adults in a setting of high dengue seroprevalence, and our results may not be generalizable to other age groups or contexts. Our study did not evaluate the stability of DBS being transported and/or stored at variable temperatures, or additional variations in the steps of the DBS elution protocol beyond the dilution step. For example, future validation studies in settings with limited access to refrigeration could consider evaluating the transport and/or storage of DBS at temperatures relevant to their context.

In conclusion, our results confirm the utility of DBS for conducting serosurveys in resource-limited settings, or when venipuncture poses a high burden to the participant (e.g., for children). Following implementation of appropriate collection and analytical methods, DBS can be used in place of serum samples analyzed by ELISA with comparatively high sensitivity, specificity, and reliability. This approach could be particularly useful for DENV seroprevalence studies in endemic regions where national surveillance is limited, such as in Brazil [26]. However, limitations remain and further advancements are still needed to develop affordable and accessible serological methods that can achieve high sensitivity to measure prior DENV infection while minimizing misclassification due to cross-reactivity from other viruses. Formal cost-benefit analyses of different approaches, including the use of DBS versus serum in ELISAs, would be insightful and help inform public health officials and researchers seeking to conduct dengue seroprevalence activities.

## Acknowledgements

We thank the participants for their contribution to the study.

## Author contributions

**Conceptualization:** Monica Zahreddine, Beatriz Parra, Mabel Carabali, Katia Charland, Valéry Ridde, Kate Zinszer.

**Data curation:** Monica Zahreddine, Beatriz Parra, Kellyanne Abreu, Danielle Malta.

**Formal analysis:** Monica Zahreddine, Katia Charland.

**Funding acquisition:** Kate Zinszer.

**Investigation:** Danielle Oliveira.

**Methodology:** Monica Zahreddine, Beatriz Parra, Danielle Oliveira, Mabel Carabali, Kellyanne Abreu.

**Project administration:** Monica Zahreddine, Beatriz Parra, Mabel Carabali, Kellyanne Abreu, Danielle Malta, Kate Zinszer.

**Resources:** Kate Zinszer.

**Supervision:** Monica Zahreddine, Danielle Malta.

**Validation:** Monica Zahreddine, Beatriz Parra, Valéry Ridde, Danielle Malta, Kate Zinszer.

**Visualization:** Katia Charland.

**Writing – original draft:** Monica Zahreddine, Laura Pierce.

**Writing – review & editing:** Monica Zahreddine, Beatriz Parra, Laura Pierce, Danielle Oliveira, Mabel Carabali, Katia Charland, Kellyanne Abreu, Valéry Ridde, Danielle Malta, Kate Zinszer.

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
