## [Decision Letter · Decision Letter 0]

9 Sep 2024

Dear Ms. Pierce,

Thank you very much for submitting your manuscript "High correlation between detection of dengue IgG from dried blood spots and serum using an indirect IgG ELISA assay: A validation study in Fortaleza, Brazil" for consideration at PLOS Neglected Tropical Diseases. As with all papers reviewed by the journal, your manuscript was reviewed by members of the editorial board and by several independent reviewers. In light of the reviews (below this email), we would like to invite the resubmission of a significantly-revised version that takes into account the reviewers' comments. 

We cannot make any decision about publication until we have seen the revised manuscript and your response to the reviewers' comments. Your revised manuscript is also likely to be sent to reviewers for further evaluation.

Sincerely,

William B Messer

Academic Editor

Michael Holbrook

Section Editor

Reviewer's Responses to Questions

**Key Review Criteria Required for Acceptance?**

**Methods**

-Are the objectives of the study clearly articulated with a clear testable hypothesis stated?

-Is the study design appropriate to address the stated objectives?

-Is the population clearly described and appropriate for the hypothesis being tested?

-Is the sample size sufficient to ensure adequate power to address the hypothesis being tested?

-Were correct statistical analysis used to support conclusions?

-Are there concerns about ethical or regulatory requirements being met?

Reviewer #1: In this manuscript, the authors evaluate the sensitivity of IgG detection extracted from dry-blood spot (DBS) by comparing it with the serum samples from young adults in a university situated in an endemic area in Brazil. The results suggested using DBS has a high agreement in IgG detection compared with using sera samples. The objective is clear and sample size is sufficient with good study design, which also include high, intermediate and low antibody sub-sample set. The following comments need to be addressed by the authors.

1. Page 6, please specify in more detail how the researchers make sure equal size of blood drops on each circle of the filter paper and what to do if the blood drop overlays to each other. If such DBS is meant to be used in limited resource area, shipping under room temperature should be tested. 

2. In the discussion, the authors described the poor sensitivity of detecting IgG by indirect IgG ELISA. Although the overall agreement between serum and DBS is quite high, the poor sensitivity of IgG ELISA could potentially bias the results when treating the equivocals from serum or DBS as negative. It is recommended that the micro-neutralization assay should be used as the gold standard and determine how equivocals should be treated in order to obtain the true sensitivity and specificity.

3. Page 7, the sentences to be more specific for the readers to understand. For example, Line 125, “undiluted DBS eluates were treated as a 1:25…..corresponding to approximately 10.5ul of serum that were eluted in 250ul of the eluent solution.” Should be …”serum that were diluted in 250ul of the diluent. Also, Line 129, the sentence “…1:2 and 1:4, to equate to 1:25… of the original blood sample” should be corrected as “of the serum” not blood. Only half of serum will be extracted from original blood sample. Also, Line 135, 100ul of each thawed serum diluted to 1:100 should be equivalent to 1:4 of DBS. The sentence should be corrected.

Reviewer #2: Objectives clearly stated. Small but sufficient sample size, appropriate stats. Evidence of ethical approval presented. Additional comments:

Abstract:

Check grammar. E.g. ‘ELISA via’ �  ‘ELISA of’ and ‘DBS limited’ �  ‘DBS’ limited’.

Author summary:

An informative summary and interpretation of results is provided.

Introduction:

Suggest not to use the word ‘drastic’, perhaps ‘significant’ instead.

Last sentence: should it read ‘optimal DBS elution protocol’ rather than ‘optimal DBS dilution’?

Methods:

First sentence check grammar.

After collection both samples were transported in refrigerated containers and the stored at -20. Why were the DBS samples refrigerated? This is contrary to one of their stated advantages in the introduction. Could this affect the external validity of your results? I.e. if DBS are transported without refrigeration in field studies, would they perform as well? �  You could address this in discussion section.

Inspection of DBS quality. How many were rejected? Were any serums rejected because of evidence of haemolysis? This is important insight into field-suitability of both sample types. Please see https://pubmed.ncbi.nlm.nih.gov/34850241/ where a significant number of blood spots were rejected.

‘Clean and sterile’ �  ‘sterile’

‘101 participants with 2 samples per person’ – does this mean the sample size requirement was actually 202? Please clarify. If it was 202 then sampling 101 individuals twice each is suboptimal, because each of the two datapoints cannot be considered independent from one another. If each individual was tested twice, please clarify whether they were sampled twice (both methods), or whether just one sample were just double tested. �  I now see from results that just 119 datapoints are analysed. Please clarify

‘We aimed to recruit 112 potential 148 participants, assuming that 10% of participants would not be eligible’ – they would not be recruited if they were not eligible.. Suggest using ‘screen’ instead of ‘recruit’ in this sentence.

Why was it decided that equivocal results would be treated as negative? Was this decided a-priori or after reviewing results? How would results be affected if the opposite decision was taken?

**Results**

-Does the analysis presented match the analysis plan?

-Are the results clearly and completely presented?

-Are the figures (Tables, Images) of sufficient quality for clarity?

Reviewer #1: 1. Page 4, Line 65, the sentence of “Seroprevalence studies commonly use DENV IgG antibodies to identify previous dengue infection…..” should be rephrased since a four-fold increase in IgG antibody titers from the sequentially collected serum samples can also be regarded as acute infection.

2. Page 5, line 91, the sentence of “The study design was cross-sectional study was conducted…” should be corrected in grammar.

3. Table 3, p-value is not 0. Please follow the scientific usage of decimals or just specify <0.05 or <0.01 or <0.001

4. 4. Page 13, line 210-212, it is better to illustrate what the acceptable range is. Based on the discussion, Line 229, if the acceptable intra or inter assay COV were acceptable under the range of 10%, then it should be stated in Line 212, the COV are all lower than 10%, instead of saying 15% as it is within 5% in fact in Table 4. Also, Table 4, the sample size of each tercils should be illustrated

Reviewer #2: Analysis presented matches plan and results are generally well presented. Additional comments:

Results:

Mean age was 22.3 years. Can you give any further details of the participants, for example their country of birth and/or whether they grew up in a dengue-endemic country (e.g. Brazil). Since they are not young children (which most serological surveillance studied focus on), it is important to try to understand the likelihood of multiple historic dengue infections (where IgG titres will be high) vs. only one (where IgG titres will be low), among the positives. Either way, this potential limitation should be picked up in the discussion: Are the findings applicable to younger populations where titres may be low? �  I now see that this is well addressed in discussion section already.

Please provide a more detailed explanation of figure 2. What are the horizontal dotted lines? Do you need to comment on the distribution of data points also – lower IVs are generally below, and higher IVs are generally above. Does this mean that DBS measurement is bias in one direction at lower concentrations, but the other direction at higher? Please explain in more detail.

Inter- and intra- assay correlation coefficients are presented. Please define these terms in more detail. Also, I am not totally clear which data these are based on. For example, do these calculations include the primary data, where all 119 were tested, or just the re-test data, where 30 positives were retested?

**Conclusions**

-Are the conclusions supported by the data presented?

-Are the limitations of analysis clearly described?

-Do the authors discuss how these data can be helpful to advance our understanding of the topic under study?

-Is public health relevance addressed?

Reviewer #1: yes

Reviewer #2: Conclusions are well supported by the data. Limitations acknowledged. Useful study in this field.

Discussion:

Fair interpretation of results. 

Some of my comments above may be better addressed in the discussion section.

In limitations, you could mention that only the dilution step in your elution protocol was varied. Is it possible that varying other steps could have yielded better results? In particular, I wonder if shaking at 300rpm overnight could have disrupted proteins and affected IgG binding.

**Editorial and Data Presentation Modifications?**

Reviewer #1: as shown in the results section

Reviewer #2: See above

**Summary and General Comments**

Reviewer #1: In the discussion, the authors described the poor sensitivity of detecting IgG by indirect IgG ELISA. Although the overall agreement between serum and DBS is quite high, the poor sensitivity of IgG ELISA could potentially bias the results when treating the equivocals from serum or DBS as negative. It is recommended that the micro-neutralization assay should be used as the gold standard and determine how equivocals should be treated in order to obtain the true sensitivity and specificity.

Reviewer #2: Thank you for inviting me to review this manuscript. Overall this is a well conducted and well presented study. The authors should be commended for doing this pragmatic research which will help others working in this field.

PLOS authors have the option to publish the peer review history of their article (what does this mean? ). If published, this will include your full peer review and any attached files.

**Do you want your identity to be public for this peer review?** For information about this choice, including consent withdrawal, please see our Privacy Policy .

Reviewer #1: Yes: Day-Yu Chao

Reviewer #2: No
---

## [Decision Letter · Decision Letter 1]

17 Dec 2024

PNTD-D-24-00702R1High correlation between detection of dengue IgG from dried blood spots and serum using an indirect IgG ELISA assay: A validation study in Fortaleza, BrazilPLOS Neglected Tropical DiseasesDear Dr. Pierce, Thank you for submitting your manuscript to PLOS Neglected Tropical Diseases. After careful consideration, we feel that it has merit but does not fully meet PLOS Neglected Tropical Diseases's publication criteria as it currently stands. Therefore, we invite you to submit a revised version of the manuscript that addresses the points raised during the review process. Please submit your revised manuscript within 30 days Jan 16 2025 11:59PM. If you will need more time than this to complete your revisions, please reply to this message or contact the journal office at plosntds@plos.org. Please include the following items when submitting your revised manuscript:

* A rebuttal letter that responds to each point raised by the editor and reviewer(s). You should upload this letter as a separate file labeled 'Response to Reviewers '. This file does not need to include responses to any formatting updates and technical items listed in the 'Journal Requirements' section below.

* A marked-up copy of your manuscript that highlights changes made to the original version. You should upload this as a separate file labeled 'Revised Manuscript with Track Changes '.

* An unmarked version of your revised paper without tracked changes. You should upload this as a separate file labeled 'Manuscript '.

If you would like to make changes to your financial disclosure, competing interests statement, or data availability statement, please make these updates within the submission form at the time of resubmission. Guidelines for resubmitting your figure files are available below the reviewer comments at the end of this letter. We look forward to receiving your revised manuscript. Kind regards,William B MesserAcademic EditorPLOS Neglected Tropical Diseases

Michael Holbrook

Section Editor

Shaden Kamhawi

co-Editor-in-Chief

Paul Brindley

co-Editor-in-Chief

**Additional Editor Comments:** Please see reviewer 2's "3 more things"**Journal Requirements:**

We note that your Data Availability Statement is currently as follows: "Requests for de-identified or anonymized data should be sent to the research ethicsboard at Université de Montréal, CERSES (email: cerses@umontreal.ca) to ensurethat data is shared in accordance with participant consent.". Please confirm at this time whether or not your submission contains all raw data required to replicate the results of your study. Authors must share the “minimal data set” for their submission. PLOS defines the minimal data set to consist of the data required to replicate all study findings reported in the article, as well as related metadata and methods (https://journals.plos.org/plosone/s/data-availability#loc-minimal-data-set-definition).

**Reviewers' comments:** Reviewer's Responses to Questions

**Key Review Criteria Required for Acceptance?**

**Methods**

-Are the objectives of the study clearly articulated with a clear testable hypothesis stated?

-Is the study design appropriate to address the stated objectives?

-Is the population clearly described and appropriate for the hypothesis being tested?

-Is the sample size sufficient to ensure adequate power to address the hypothesis being tested?

-Were correct statistical analysis used to support conclusions?

-Are there concerns about ethical or regulatory requirements being met?

Reviewer #1: The authors have properly addressed the key points and revised the manuscript

Reviewer #2: (No Response)

**Results**

-Does the analysis presented match the analysis plan?

-Are the results clearly and completely presented?

-Are the figures (Tables, Images) of sufficient quality for clarity?

Reviewer #1: yes

Reviewer #2: Thank you for addressing my earlier comments and revising this manuscript. It is a nice piece of work.

3 more things:

1. I would prefer percentages in table 2 to be 'row percentages' rather than 'column percentages'. I believe this would make it more immediately interpretable. Also add a 'total' column on the right. Not an essential change.

2. It is still not clear to me what each of the horizontal dotted lines are, in Figure 4. I think further explanation is needed, or they could be removed.

3. It is clear from Figure 3, Figure 4, and from your analysis of discrepant results that DBS is likely giving higher IVs than serum at the low end of the assay range (i.e. below 1.5, which includes the serological cutoff at 0.9-1.1). This is mentioned in the discussion BUT I don't think the potential impact of this observation is adequately discussed yet. In this study of adults, IGG was mostly detected strongly (high IV) or not detected (low IV). There were very few results near to the cutoff. Therefore, the authors observed very good agreement between DBS and serum. However, if there were more samples closer to the cutoff, this likely would not have been the case. Children, who for many reasons are generally targeted in dengue serosurveys (and who are specifically mentioned in the intro to this paper) may have a higher proportion of 'monotypic' antibody responses (i.e. after only one previous infection) and may have lower concentrations of IgG. In this group, DBS could over-call seropositivity more (compared to adults).

Were there any experiments using DBS elude at 1:8? It looks as if further dilution of the elute may have resulted in IV equivalence when it was 0.9-1.1 , which I would argue is the 'optimal' target.

Anyway, I do think this needs to be set out a bit more clearly, as any subsequent researchers/studies which use this DBS protocol would need to be careful with any DBS results which are around the cutoff (and crucially what proportion of the results are near the cutoff).

**Conclusions**

-Are the conclusions supported by the data presented?

-Are the limitations of analysis clearly described?

-Do the authors discuss how these data can be helpful to advance our understanding of the topic under study?

-Is public health relevance addressed?

Reviewer #1: Yes

Reviewer #2: (No Response)

**Editorial and Data Presentation Modifications?**

Reviewer #1: The authors have properly addressed the key points and revised the manuscript

Reviewer #2: (No Response)

**Summary and General Comments**

Reviewer #1: (No Response)

Reviewer #2: (No Response)

PLOS authors have the option to publish the peer review history of their article (what does this mean? ). If published, this will include your full peer review and any attached files.

**Do you want your identity to be public for this peer review?** For information about this choice, including consent withdrawal, please see our Privacy Policy .

Reviewer #1: **Yes: ** Day-Yu Chao

Reviewer #2: **Yes: ** Paul Arkell

**Figure resubmission:** While revising your submission, please upload your figure files to the Preflight Analysis and Conversion Engine (PACE) digital diagnostic tool, https://pacev2.apexcovantage.com/. PACE helps ensure that figures meet PLOS requirements. To use PACE, you must first register as a user. Registration is free. Then, login and navigate to the UPLOAD tab, where you will find detailed instructions on how to use the tool. If you encounter any issues or have any questions when using PACE, please email PLOS at figures@plos.org. Please note that Supporting Information files do not need this step. If there are other versions of figure files still present in your submission file inventory at resubmission, please replace them with the PACE-processed versions.**Reproducibility:** To enhance the reproducibility of your results, we recommend that authors of applicable studies deposit laboratory protocols in protocols.io, where a protocol can be assigned its own identifier (DOI) such that it can be cited independently in the future. Additionally, PLOS ONE offers an option to publish peer-reviewed clinical study protocols. Read more information on sharing protocols at https://plos.org/protocols?utm_medium=editorial-email&utm_source=authorletters&utm_campaign=protocols

---

## [Editor Report · Decision Letter 2]

21 Jan 2025

PNTD-D-24-00702R2

High correlation between detection of dengue IgG from dried blood spots and serum using an indirect IgG ELISA assay: A validation study in Fortaleza, Brazil

Dear Dr. Pierce,

Thank you for submitting your manuscript to PLOS Neglected Tropical Diseases. After careful consideration, we feel that it has merit but does not fully meet PLOS Neglected Tropical Diseases's publication criteria as it currently stands. Therefore, we invite you to submit a revised version of the manuscript that addresses the points raised during the review process by one of the reviewers:

1. I would prefer percentages in table 2 to be 'row percentages' rather than 'column percentages'. I believe this would make it more immediately interpretable. Also add a 'total' column on the right. Not an essential change.

2. It is still not clear to me what each of the horizontal dotted lines are, in Figure 4. I think further explanation is needed, or they could be removed.

3. It is clear from Figure 3, Figure 4, and from your analysis of discrepant results that DBS is likely giving higher IVs than serum at the low end of the assay range (i.e. below 1.5, which includes the serological cutoff at 0.9-1.1). This is mentioned in the discussion BUT I don't think the potential impact of this observation is adequately discussed yet. In this study of adults, IGG was mostly detected strongly (high IV) or not detected (low IV). There were very few results near to the cutoff. Therefore, the authors observed very good agreement between DBS and serum. However, if there were more samples closer to the cutoff, this likely would not have been the case. Children, who for many reasons are generally targeted in dengue serosurveys (and who are specifically mentioned in the intro to this paper) may have a higher proportion of 'monotypic' antibody responses (i.e. after only one previous infection) and may have lower concentrations of IgG. In this group, DBS could over-call seropositivity more (compared to adults).

Were there any experiments using DBS elude at 1:8? It looks as if further dilution of the elute may have resulted in IV equivalence when it was 0.9-1.1 , which I would argue is the 'optimal' target.

Anyway, I do think this needs to be set out a bit more clearly, as any subsequent researchers/studies which use this DBS protocol would need to be careful with any DBS results which are around the cutoff (and crucially what proportion of the results are near the cutoff).

Please submit your revised manuscript within 30 days Feb 20 2025 11:59PM. If you will need more time than this to complete your revisions, please reply to this message or contact the journal office at plosntds@plos.org. Please include the following items when submitting your revised manuscript:

* A rebuttal letter that responds to each point raised by the editor and reviewer(s). You should upload this letter as a separate file labeled 'Response to Reviewers '. This file does not need to include responses to any formatting updates and technical items listed in the 'Journal Requirements' section below.

* A marked-up copy of your manuscript that highlights changes made to the original version. You should upload this as a separate file labeled 'Revised Manuscript with Track Changes '.

* An unmarked version of your revised paper without tracked changes. You should upload this as a separate file labeled 'Manuscript '.

We look forward to receiving your revised manuscript.

Kind regards,

William B Messer

Academic Editor

Michael Holbrook

Section Editor

Shaden Kamhawi

co-Editor-in-Chief

Paul Brindley

co-Editor-in-Chief

**Journal Requirements:**

1) We note that your Data Availability Statement is currently as follows: "Requests for de-identified or anonymized data should be sent to the research ethicsboard at Université de Montréal, CERSES (email: cerses@umontreal.ca) to ensurethat data is shared in accordance with participant consent since data contain potentially sensitive personal health information. Once the data access request to the research ethics board has been approved, all de-identified or anonymized data used to replicate this study's findings will be made available by the corresponding author.". Please confirm at this time whether or not your submission contains all raw data required to replicate the results of your study. Authors must share the “minimal data set” for their submission. PLOS defines the minimal data set to consist of the data required to replicate all study findings reported in the article, as well as related metadata and methods (https://journals.plos.org/plosone/s/data-availability#loc-minimal-data-set-definition).

- The points extracted from images for analysis..

**Reviewers' comments:** **Figure resubmission:** While revising your submission, please upload your figure files to the Preflight Analysis and Conversion Engine (PACE) digital diagnostic tool, https://pacev2.apexcovantage.com/. PACE helps ensure that figures meet PLOS requirements. To use PACE, you must first register as a user. Registration is free. Then, login and navigate to the UPLOAD tab, where you will find detailed instructions on how to use the tool. If you encounter any issues or have any questions when using PACE, please email PLOS at figures@plos.org. Please note that Supporting Information files do not need this step. If there are other versions of figure files still present in your submission file inventory at resubmission, please replace them with the PACE-processed versions.
---

## [Editor Report · Decision Letter 3]

29 Jan 2025

Dear Ms. Pierce,

We are pleased to inform you that your manuscript 'High correlation between detection of dengue IgG from dried blood spots and serum using an indirect IgG ELISA assay: A validation study in Fortaleza, Brazil' has been provisionally accepted for publication in PLOS Neglected Tropical Diseases.

Best regards,

William B Messer

Academic Editor

Michael Holbrook

Section Editor

Shaden Kamhawi

co-Editor-in-Chief

Paul Brindley

co-Editor-in-Chief

---

## [Editor Report · Acceptance letter]

Dear Ms. Pierce,

We are delighted to inform you that your manuscript, "High correlation between detection of dengue IgG from dried blood spots and serum using an indirect IgG ELISA assay: A validation study in Fortaleza, Brazil," has been formally accepted for publication in PLOS Neglected Tropical Diseases.

Best regards,

Shaden Kamhawi

co-Editor-in-Chief

Paul Brindley

co-Editor-in-Chief
